# Diagnosing Ovarian Cancer on MRI: A Preliminary Study Comparing Deep Learning and Radiologist Assessments

**DOI:** 10.3390/cancers14040987

**Published:** 2022-02-16

**Authors:** Tsukasa Saida, Kensaku Mori, Sodai Hoshiai, Masafumi Sakai, Aiko Urushibara, Toshitaka Ishiguro, Manabu Minami, Toyomi Satoh, Takahito Nakajima

**Affiliations:** 1Department of Radiology, Faculty of Medicine, University of Tsukuba, Tsukuba 305-8575, Japan; saida_sasaki_tsukasa@md.tsukuba.ac.jp (T.S.); moriken@md.tsukuba.ac.jp (K.M.); hoshiai@md.tsukuba.ac.jp (S.H.); msakai-kgs@umin.ac.jp (M.S.); urushibara-kgw@umin.ac.jp (A.U.); toishiguro@md.tsukuba.ac.jp (T.I.); mminami@md.tsukuba.ac.jp (M.M.); 2Department of Obstetrics and Gynecology, Faculty of Medicine, University of Tsukuba, Tsukuba 305-8575, Japan; toyomi-s@md.tsukuba.ac.jp

**Keywords:** ovary, carcinoma, artificial intelligence, convolutional neural network, magnetic resonance imaging

## Abstract

**Simple Summary:**

As a preliminary experiment to explore the possibility of clinical application as a future reading assist, we present CNNs for the diagnosis of ovarian carcinomas and borderline tumors on MRI, including T2WI, DWI, ADC map, and CE-T1WI, and compare their diagnostic performance with interpretations by experienced radiologists. CNNs were trained using 1798 images from 146 patients and 1865 images from 219 patients with malignant tumors, including borderline tumors, and non-malignant lesions, respectively, for each MRI sequence and tested with 48 and 52 images of patients with malignant and non-malignant lesions. The CNN of each sequence had a sensitivity of 0.77–0.85, specificity of 0.77–0.92, accuracy of 0.81–0.87, and an AUC of 0.83–0.89, demonstrating diagnostic performances that were non-inferior to those of experienced radiologists, and the CNN showed the highest diagnostic performance on the ADC map for each sequence (specificity = 0.85; sensitivity = 0.77; accuracy = 0.81; AUC = 0.89).

**Abstract:**

Background: This study aimed to compare deep learning with radiologists’ assessments for diagnosing ovarian carcinoma using MRI. Methods: This retrospective study included 194 patients with pathologically confirmed ovarian carcinomas or borderline tumors and 271 patients with non-malignant lesions who underwent MRI between January 2015 and December 2020. T2WI, DWI, ADC map, and fat-saturated contrast-enhanced T1WI were used for the analysis. A deep learning model based on a convolutional neural network (CNN) was trained using 1798 images from 146 patients with malignant tumors and 1865 images from 219 patients with non-malignant lesions for each sequence, and we tested with 48 and 52 images of patients with malignant and non-malignant lesions, respectively. The sensitivity, specificity, accuracy, and AUC were compared between the CNN and interpretations of three experienced radiologists. Results: The CNN of each sequence had a sensitivity of 0.77–0.85, specificity of 0.77–0.92, accuracy of 0.81–0.87, and an AUC of 0.83–0.89, and it achieved a diagnostic performance equivalent to the radiologists. The CNN showed the highest diagnostic performance on the ADC map among all sequences (specificity = 0.85; sensitivity = 0.77; accuracy = 0.81; AUC = 0.89). Conclusion: The CNNs provided a diagnostic performance that was non-inferior to the radiologists for diagnosing ovarian carcinomas on MRI.

## 1. Introduction

Ovarian cancer is the eighth most common cancer diagnosis and cause of cancer death in women; there were approximately 313,959 cases and 207,252 deaths worldwide in 2020 [1]. Most women presenting with ovarian cancer are asymptomatic; even if they are symptomatic, the symptoms are nonspecific and difficult to screen. For this, patients are often already at an advanced stage at the time of diagnosis, making ovarian cancer the “silent killer” [2]. Transvaginal ultrasonography (US) is performed to screen for ovarian cancer with a clinically suspected adnexal mass with 82–92% accuracy [3]. However, approximately 5–20% of adnexal masses remain uncharacterized following US. For these indeterminate masses, although short-term follow up is an option, magnetic resonance imaging (MRI) is a problem-solving tool [4].

MRI is a better radiologic method for differentiating between malignant and benign ovarian tumors [4,5,6]. However, MRI may not be suitable for patients with massive ascites. Among the MRI sequences, T2-weighted imaging (T2WI) has high soft-tissue contrast resolution and is the most basic sequence for anatomical evaluation. Diffusion-weighted imaging (DWI) and apparent diffusion coefficient (ADC) values are sensitive and relatively specific methods for differentiating malignant from benign tumors [7,8]. Gadolinium-enhanced MRI is also highly accurate for detecting malignant ovarian tumors [6,9]. In 2013, Thomassin-Naggara et al. published the adnexal MRI scoring system (i.e., O–RADS MRI) for characterization of adnexal masses that were indeterminate on US. O–RADS MRI scores range from 0 to 5, with 0 indicating an incomplete evaluation; 1 indicating a normal ovary; 2 indicating a pure cystic mass, pure fatty mass, or pure endometriotic cyst that is almost certainly benign; 3 indicating a low risk: a cystic tumor with no enhancing solid tissue or a tumor containing solid tissue with a low-risk time–intensity curve on dynamic contrast study; 4 indicating intermediate risk: a tumor containing solid tissue with an intermediate-risk time–intensity curve; 5 indicating high risk: a tumor containing solid tissue with a high-risk time–intensity curve. This scoring system has a sensitivity of 93.5% and a specificity of 96.6% for stratifying the risk of malignancy in adnexal masses and is currently widely used [10]. However, the differentiation between adenomas and malignant tumors, especially borderline tumors, based on MRI is often problematic, and several studies have reported that a cystic tumor interpreted as benign, even on an MRI, required further examination to exclude the possibility of it being a borderline tumor [5,11].

Convolutional neural networks (CNNs) are a class of deep learning models that combine imaging filters with artificial neural networks through a series of successive linear and nonlinear layers. It is considered a promising tool for diagnostic imaging and, recently, several CNNs for diagnostic imaging have been constructed and achieved excellent performance in image classification using radiography, US, CT, and MRI [12], and CNN could be used as a reading assist for screening and scrutiny of ovarian cancer by MRI.

As a preliminary experiment to explore their possibility of clinical application as a future reading assist, we present CNNs for the diagnosis of ovarian carcinomas and borderline tumors on MRI, including T2WI, DWI, ADC map, and fat-saturated contrast-enhanced T1-weighted imaging (CE-T1WI), and compared their diagnostic performance with interpretations by experienced radiologists.

## 2. Materials and Methods

### 2.1. Patients

This retrospective study was approved by our local institutional review board, which waived the need for written informed consent (approval number: R02-112). The inclusion criteria were as follows: (a) aged above 20 years for ethical reasons; (b) pelvic MRI scan obtained as per the protocol followed at our hospital between January 2015 and December 2020; (c) pathologically proven malignant epithelial tumors (i.e., carcinomas) or borderline tumors of the ovary for the malignant group; (d) pathologically proven or clinically apparent benign lesions in the non-malignant group. The exclusion criteria were as follows: (a) malignant tumors in the pelvis other than the ovary; (b) past history of surgery of the uterus or ovaries other than caesarean section, chemotherapy, or radiation therapy of the pelvis; (c) malignant ovarian epithelial tumors mixed with non-epithelial components. A flowchart of the patient selection process is shown in Figure 1.

Tumor stage was comprehensively determined based on the International Federation of Gynecology and Obstetrics (FIGO) 2014 classification using pretreatment MRI, CT, surgical, and pathological findings. For operated cases, tumor type was determined by pathological diagnosis, and mixed carcinomas were classified according to the most dominant histological type.

### 2.2. MRI Acquisition

MRI was performed using 3 T or 1.5 T equipment (Ingenia^®^, Achieva^®^; Philips Medical Systems, Amsterdam, Netherlands). The protocol employed to obtain the image along the uterine short axis included T2WI, DWI with a b value of 0 and 1000, and CE-T1WI with gadopentetate dimeglumine 5 mmol (Magnevist^®^ 0.5 M or Gadovist^®^ 1.0 M; Bayer, Wuppertal, Germany). The bolus intravenous contrast injection rate was 4 mL (2 mmol)/s (in case of Gadovist, diluted with saline solution and injected at 4 mL/s). Further details of these parameters are provided in Table 1.

### 2.3. Data Set

To detect the most suitable sequence for the CNN in malignant/non-malignant tumor discrimination, a total of 4 sequences were obtained: oblique axial T2WI, oblique axial DWI, oblique axial ADC map, and oblique axial CE-T1WI.

To create a data set, only the slices in which the tumor was visualized were extracted from the ovarian tumor, both malignant and non-malignant, based on the consensus of the two radiologists (T.S., A.U.). For benign lesions other than ovarian tumors, the slices in which the uterus and ovaries were visualized were extracted. The same cross-section was extracted for all sequences.

A total of 465 patients were randomly assigned to the training and testing groups. In the training phase, 3663 images for each sequence from 365 patients (1798 images from 146 patients in the malignant group and 1865 images from 219 patients in the non-malignant group) were used. In the test phase, based on the consensus of the two radiologists (T.S., A.U.), one slice indicating a solid component, a central slice for each ovarian tumor containing no solid components, or an image showing a normal caudal ovary in the absence of an ovarian tumor was selected. According to O–RADS MRI, solid tissue was defined as a lesion component that conformed to one of these morphologies: papillary projection, mural nodule, irregular septation/wall, or other larger solid portions. Then, for each sequence, 100 images from 100 patients (48 images of 48 malignant tumors and 52 images of 52 non-malignant lesions) were used.

In this study, the software used was unable to handle the Digital Imaging and Communications in Medicine (DICOM) images; therefore, the DICOM images were converted into Joint Photographic Experts Group (JPEG) images that employ a lossless compression method using a Centricity Universal Viewer (GE Healthcare, Chicago, IL, USA). Next, the margins containing the patient information were automatically trimmed and resized to 240 × 240 pixels using XnConvert (Gougelet Pierre-Emmanuel, Reims, France).

### 2.4. Deep Learning Using CNNs

Deep learning was performed on a deep station entry (UEI, Tokyo, Japan) with a GeForce RTX 2080Ti graphics processing unit (NVIDIA, Delaware, CA, USA), a Core i7-8700 central processing unit (Intel, Santa Clara, CA, USA), and the deep learning software Deep Analyzer (GHELIA, Tokyo, Japan).

The conditions were optimized based on the ablation and comparative studies of the previous research as follows: a CNN with Xception architecture was used for deep learning. Xception is characterized as depth-wise separable convolutions that enable the use of model parameters more efficiently than the previous CNN architecture [13]. ImageNet, which comprises natural images as pre-trained data, was used for the pre-training [14]; the optimizer algorithm = Adam (learning rate = 0.0001, β1 = 0.9, β2 = 0.999, eps = le-7, decay = 0, and AMSGrad = false); horizontal flip, rotation (4.5°), shearing (0.05), and zooming (0.05) were also automatically used as data augmentation techniques. The validation ratio (validation/training) was set at 0.1 or 0.2. Fifty, 100, and 200 epochs were used for the training. The batch size was automatically selected by Deep Analyzer to fit into the graphics processing unit memory.

### 2.5. Radiologist Interpretation

Three experienced radiologists (K.M., S.H., and M.S.), with 26, 12, and 8 years of experience in interpreting pelvic MRIs independently, reviewed the 100 test images of each sequence from 100 patients in random order. They evaluated each image by assigning confidence levels to the diagnosis of malignant tumors, including borderline tumors, using a 6 point scale (1.0, definitely malignant; 0.8, probably malignant; 0.6, possibly malignant; 0.4, possibly benign; 0.2, probably benign; 0, definitely benign). The radiologists were blinded to the pathological and clinical findings. A duration of one week or more was observed between each sequence interpretation.

### 2.6. Statistical Analysis

The age and histological type for each group were compared using the Mann–Whitney U test and the chi-square test.

The test data set was used to calculate the sensitivity, specificity, and accuracy of diagnosing malignant tumors. With the radiologists, 1.0–0.6 was treated as malignant, while 0.4–0.0 was treated as non-malignant. In the CNN, the classification into malignant and non-malignant groups was output as a continuous number from 1 to 0: 1.00–0.50 was considered malignant, while 0.49–0 was considered non-malignant.

Receiver operating characteristic (ROC) curve analysis was performed to assess diagnostic performance. Moreover, the area under the receiver operating characteristic curve (AUC) was compared between the CNN and the radiologists with their 95% confidence intervals, and significant differences were estimated [15].

Inter-observer agreement for the two choices of malignancy was also assessed using kappa (κ) statistics. The κ-statistic interpreted the agreement as follows: 1.00–0.81, almost perfect; 0.80–0.61, substantial; 0.60–0.41, moderate; 0.40–0.21, fair; 0.20–0, slight; less than 0, none [16]. All statistical analyses were performed using SPSS software (SPSS Statistics 27.0; IBM, New York, NY, USA). Statistical significance was set at *p* < 0.05.

## 3. Results

A total of 465 women (mean age, 50 years; age range, 20–90 years) were evaluated across the data sets. Table 2 shows the patients’ characteristics and pathological types of malignant and non-malignant lesions. The high percentage of stage Ⅰ malignant tumors was due in part to the inclusion of borderline tumors. In the malignant group, serous carcinoma, clear cell carcinoma, and mucinous tumor (i.e., carcinoma and borderline tumor) had high ratios. Although patients in the malignant group were significantly older than those in the non-malignant group (*p* < 0.001), there was no significant difference between the training and testing data in patient age or histological tumor type. In the malignant group, 116 (training, 88; testing, 28) patients were scanned at 1.5 T, and 78 (training, 58; testing, 20) patients were scanned at 3 T. In the non-malignant group, 114 (training, 91; testing, 23) patients were scanned at 1.5 T, and 157 (training, 128; testing, 29) patients were scanned at 3 T. All tumors in the malignant group were confirmed pathologically. Only 57 (training, 48; testing, 9) lesions in the non-malignant group were not pathologically confirmed but were clinically apparent benign lesions including imaging findings such as small myoma and nabothian cyst.

For the selection of the validation ratio and epochs, among the CNNs with a sensitivity and specificity above 0.75, a model with a validation ratio of 0.1 and 100 epochs was adopted for T2WI, a model with a validation ratio of 0.2 and 50 epochs was adopted for DWI, and a model with a validation ratio of 0.1 and 50 epochs was adopted for the ADC map (Figure 2), while a model with a validation ratio of 0.2 and 200 epochs was adopted for CE-T1WI due to the fact of their high diagnostic performance.

Table 3 lists the diagnostic performance of the CNNs versus the radiologists. The ROC curves comparing them are shown in Figure 3, and the sensitivity, specificity, and accuracy of the CNN for each sequence were comparable to those of the three radiologists. No significant difference was observed between the CNN and the three radiologists except for significantly higher AUCs on the DWI and ADC map and a significantly lower AUC on the T2WI for the CNN than for reader 2. The CNN showed the highest diagnostic performance on the ADC map with an AUC of 0.89.

The CNN correctly diagnosed 28 of 37 carcinomas and 9 of 11 borderline tumors on T2WI; 31 of 37 carcinomas and 10 of 11 borderline tumors on the DWI and ADC map; 31 of 37 carcinomas and 8 of 11 borderline tumors on the CE-T1WI as malignant tumors. There was one case each in which the CNN showed false negative and false positive in all sequences. Reader 1 had three cases with false negatives, while readers 2 and 3 had no cases with false negatives or false positives in all sequences.

Figure 4, Figure 5 and Figure 6 show the test images of three cases in this study with different interpretations by the CNN and radiologists including the confidence value. Figure 4 and Figure 5 show the cases in which the CNN was able to make the correct diagnosis, although the radiologists had a high rate of false-negative diagnoses. Figure 4 shows a tumor with a slight signal difference between the solid components and the background on the T2WI, DWI, and ADC map. Figure 5 shows a non-malignant tumor containing components that were difficult to distinguish between solid components and mucus on the T2WI; however, the CNN diagnosed it as a non-malignant tumor. Figure 6 shows a case in which the CNN and the radiologists showed false negatives on the sequences other than the DWI, and as for the radiologists, it was assumed that the misdiagnosis was due to the difficulty in distinguishing the solid component of the ovarian tumor from the intestinal tract.

Table 4 shows the inter-observer agreement between the CNN and the three radiologists’ assessments. The κs between the CNN and radiologists was 0.17–0.63, varying widely and less consistent than those among the radiologists. The κs between radiologists were low on the ADC map and varied widely.

## 4. Discussion

This preliminary study presented CNNs for diagnosing ovarian carcinomas, including borderline tumors, using several MRI sequences, demonstrating that the diagnostic performance was not inferior to that of the experienced radiologists, under the limited conditions.

Few studies have used deep learning in the field of gynecological imaging. Urushibara et al. recently constructed a CNN that showed good diagnostic performance for identifying the presence of cervical cancer on T2WI [17]. Aramendía et al. developed a CAD technique for US images that was able to discriminate between malignant and benign adnexal masses based on a texture analysis of 145 patients [18]. Jian et al. and Li et al. constructed an MRI-based texture analysis model to distinguish between type I and type II epithelial ovarian cancers and borderline and malignant epithelial ovarian tumors based on T2WI+DWI+ADC and CE-T1WI+T2WI [19,20]. Wang et al. developed a CNN that distinguishes benign from malignant ovarian on T2WI, CE-T1WI, and clinical variables [21]. To the best of our knowledge, this is the first image classification study of the diagnosis of ovarian tumors using MRI images including DWI and ADC map via deep learning. It is also noteworthy that we used entire pelvic images, not just the cropped images of the ovarian lesion alone.

Epithelial ovarian tumors can be classified into serous, mucinous, endometrioid, clear cell, Brenner, seromucinous, and undifferentiated types. Furthermore, each is subdivided into benign, borderline, or malignant [22]. One of the purposes of MRI is to distinguish malignant tumors from benign lesions and infer their histology; however, the imaging findings of ovarian tumors differ greatly depending on their histological type. For example, high-grade serous carcinoma typically tends to appear as small, bilateral masses often accompanied by peritoneal dissemination [23]; mucinous tumors can be very large, and the signal intensity of the mucinous content is variable [24]; endometrioid, clear cell, and seromucinous tumors are associated with endometriosis [23,24,25]. Moreover, distinguishing between borderline tumors and adenomas is often difficult, even for experienced radiologists [5,26,27]. However, in order to explore the potential clinical applications of CNN, it was considered essential to distinguish borderline tumors with potentially malignant behavior from benign lesions; therefore, borderline tumors were included in the malignant group in this study.

The image interpretation of ovarian tumors is complicated as mentioned above. However, although our CNNs included less than 500 cases and the images used were not cropped images of the ovarian lesions, they showed diagnostic performances equivalent to those of experienced radiologists in our study. In addition, there was no inferiority in diagnostic performance even for borderline tumors, which are generally difficult to diagnose as benign or malignant. The fact that the CNN showed the highest diagnostic performance with the ADC map is consistent with a previous study of the diagnosis of prostate cancer [28], indicating that the ADC map is a suitable image for CNN-based diagnosis.

The inter-observer agreement between the CNNs and the radiologists was lower than that between the radiologists; however, the agreement between the radiologists was not high. Radiologists usually make a comprehensive judgment by referring to multiple sequences; therefore, making a diagnosis using only a single sequence is difficult. For example, in the case of ovarian endometriosis, the contrast between the background of the cyst content showing T2 shading and the solid components is poor, and the solid components might be noticed only on CE-T1WI. For endometriosis showing hyperintensity on T1WI, it can be difficult to recognize the contrast enhancement of the solid components even on CE-T1WI, and the use of a subtracted image is required. Therefore, the use of a single sequence without multiple sequences may have undermined the overall agreement among radiologists.

Our study had several limitations. First, each sequence was evaluated individually, which is quite different from clinical practice in which all sequences are referenced and comprehensively diagnosed and did not meet the definition of O-RADS MRI. It also differed from O-RADS in that it was scored by confidence level rather than malignancy for comparison with the diagnosis made by the CNN. In addition, the lack of interpretation in the series images is believed to be another reason why the radiologists’ performance was worse than in past O-RADS reports. Using a combination of several sequences may show superior diagnostic performance to the current models as reported for the CNN of prostate cancer using fused images [28,29], and the fact that there was only one false positive and negative in all sequences in this study is a result that can be expected to improve the diagnostic performance of combination imaging in the future; however, our unpublished data indicated that this case size cannot be expected to improve interpretation of the CNN with use of a combined image set, and more training images may be required to achieve higher diagnostic performance with combination images. Second, because the test images were intentionally selected, selection bias was unavoidable. Third, to avoid study complexity, we targeted only epithelial tumors, and our CNNs might not be able to diagnose other types of malignant ovarian tumors. Fourth, although pathological findings were used as reference, it was sometimes difficult to distinguish between borderline tumors and adenomas, even pathologically. The non-malignant group included lesions that were not pathologically confirmed; however, we thought that it was important to distinguish between malignant tumors and benign lesions that were not indicated for treatment. Fifth, this study included rare tumors, such as seromucinous tumors, and future research will be required to demonstrate that the model achieves satisfactory performance for these rare tumor types. In addition, the following can be considered future improvements: deep learning of all images acquired from all sequences based on DICOM data, while also incorporating clinical information such as age and tumor markers; transferring learning using medical image training [30]; using images taken with other MRI systems for deep learning.

## 5. Conclusions

Although diagnostic imaging of ovarian tumors is complex, deep learning has shown good diagnostic performance for diagnosing ovarian carcinomas, including borderline tumors, on MRIs under the limited conditions of this study.

## Figures and Tables

**Figure 1 cancers-14-00987-f001:**
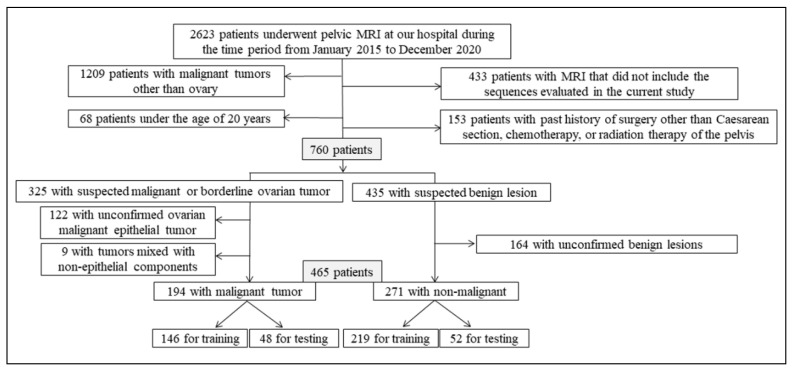
Flowchart for the patient selection process.

**Figure 2 cancers-14-00987-f002:**
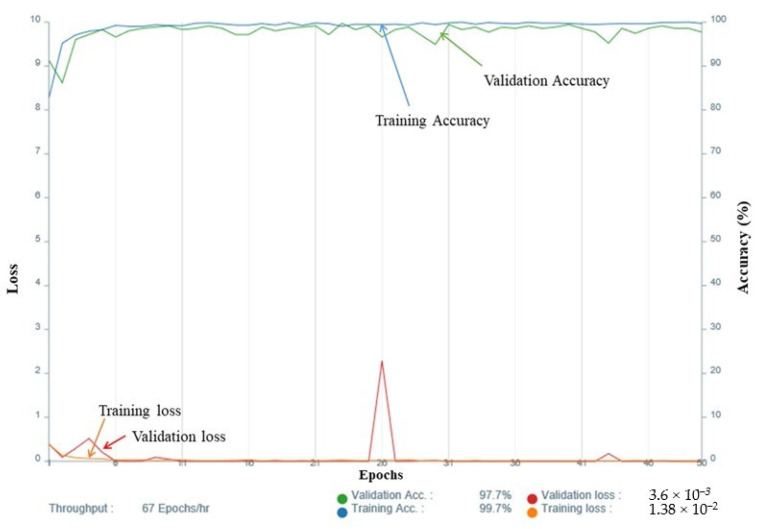
Accuracy and loss of the training data (apparent diffusion coefficient map with a validation ratio of 0.1 and 50 epochs).

**Figure 3 cancers-14-00987-f003:**
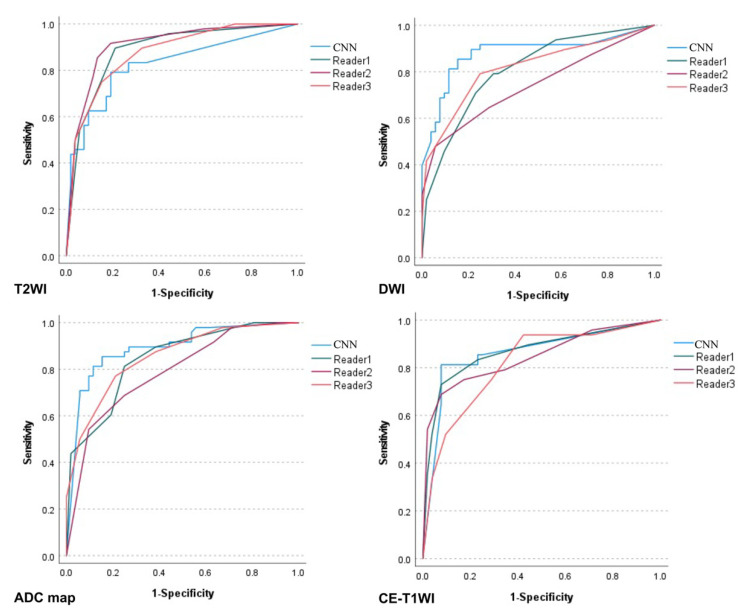
Receiver operating characteristic curves for the performance of the convolutional neural network versus the radiologists’ performances.

**Figure 4 cancers-14-00987-f004:**
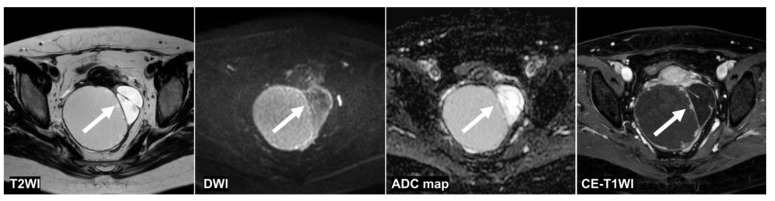
A 51 year old woman with a seromucinous borderline tumor. Only the CNN could diagnose malignant tumors on the T2WI and the DWI (the CNN confidence value: malignant = 98.5% on T2WI; malignant = 99.9% on DWI). The CNN and reader 2 could diagnose malignant tumors on the ADC map (the CNN confidence value: malignant = 82.1%). On the other hand, the CNN and all radiologists could diagnose malignant tumors on the CE-T1WI (the CNN confidence value: malignant = 99.9%). This case was a typical image of seromucinous borderline or serous borderline tumor. There was almost no contrast between the papillary projections (arrow) showing hyperintensities on the T2WI and the contents of the cyst, and it was difficult to identify them, other than CE-T1WI, for the radiologists. ADC: apparent diffusion coefficient; CE-T1W1: contrast-enhanced T1-weighted imaging; CNN: convolutional neural network; DWI: diffusion-weighted imaging.

**Figure 5 cancers-14-00987-f005:**
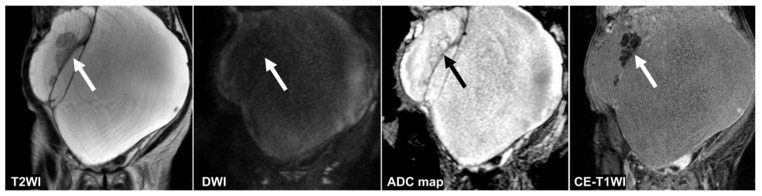
A 95 year old woman with mucinous cystadenoma. Only the CNN could diagnose non-malignant tumors on the T2WI (CNN confidence value; malignant = 0.0%). The CNN and all radiologists could diagnose non-malignant tumors on the DWI (CNN confidence value; malignant = 0.0%). The CNNs and only one reader could diagnose non-malignant tumors on the ADC map and the CE-T1WI (CNN confidence value: malignant = 0.0% on the ADC map and the DWI). Mucus (arrow) showed intermediate signal intensities and was indistinguishable from solid components on the T2WI. The septum (arrow) appeared dense on the ADC map and the CE-T1WI, making it difficult to distinguish it from the borderline tumor. ADC: apparent diffusion coefficient; CE-T1WI: contrast-enhanced T1-weighted imaging; CNN: convolutional neural network; DWI: diffusion-weighted imaging; T2WI: T2-weighted imaging.

**Figure 6 cancers-14-00987-f006:**
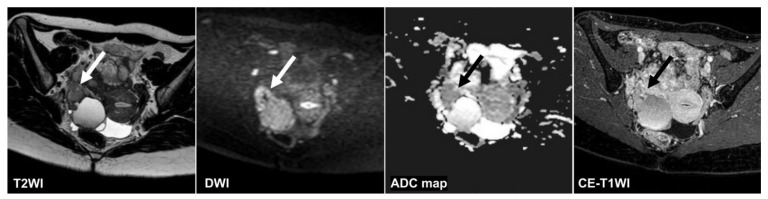
A 28 year old woman with high-grade serous carcinoma. None of the CNNs or the three radiologists could diagnose malignant tumors on the T2WI and the ADC map (CNN confidence value: malignant = 0.0% on the T2WI; malignant = 1.5% on the ADC map). Only reader 3 could diagnose a malignant tumor on the CE-T1WI (CNN confidence value: malignant = 0.0%). In contrast, the CNN and all radiologists could diagnose malignant tumors on the DWI (the CNN confidence value; malignant = 99.9%). It seemed it was difficult to distinguish the tumor (arrow) from the intestinal tract. ADC: apparent diffusion coefficient; CE-T1WI: contrast-enhanced T1-weighted imaging; CNN: convolutional neural network; DWI: diffusion-weighted imaging; T2WI: T2-weighted imaging.

**Table 1 cancers-14-00987-t001:** Acquisition parameters of magnetic resonance imaging.

Sequence	Type	Repetition Time/Echo Time (ms)	Flip Angle (Degree)	Slice/Gap (mm)	Field of View (mm)	Matrix
T2WI	2D Turbo-spin echo	1400–6013/10–110	90	3–5/0.3–1	260–380	512 × 512–704 × 704
DWI	Echo planar imaging	4068–7500/70–79	90	3–5/0–1	260–380	224 × 224–352 × 352
CE-T1WI	3D Gradient echo spectral pre-saturation with inversion recovery	4–5/2	10–15	2.2–3.3/0–1.6	260–380	352 × 352–704 × 704

CE-T1WI: contrast-enhanced fat-saturated T1-weighted imaging; DWI: diffusion-weighted imaging; T2WI: T2-weighted imaging.

**Table 2 cancers-14-00987-t002:** Patient and lesion characteristics.

Variable	Training Data			Testing Data		
	Malignant Group	Non-Malignant Group	All	Malignant Group	Non-Malignant Group	All
Patients (*n*)	146	219	365	48	52	100
Images (slices)	1798	1865	3663	48	52	100
Age						
Mean ± standard deviation (y)	55 ± 14	47 ± 13	50 ± 14	55 ± 14	45 ± 14	50 ± 15
Range (y)	20–87	21–86	20–87	22–76	20–90	20–90
Tumor stage of malignant group (*n*) (I/Ⅱ/Ⅲ/Ⅳ)	83/17/34/22			28/3/14/3		
Tumor type of malignant group (*n*)						
Serous tumor (HGSC/LGSC/BOT)	39/1/5			14/0/1		
Clear cell tumor (carcinoma/BOT)	40/0			13/0		
Mucinous tumor (carcinoma/BOT)	15/18			4/6		
Endometrioid tumor (carcinoma/BOT)	14/4			5/2		
Seromucinous tumor (carcinoma/BOT)	4/6			1/2		
Tumor type of non-malignant group (*n*)						
Serous tumor (cystadenoma/adenofibroma)		15/6			6/1	
Mucinous tumor (cystadenoma/adenofibroma)		34/1			10/0	
Seromucinous cystadenoma		2			1	
Endometriosis		28			7	
Mature teratoma		16			6	
Leiomyoma		56			10	
Uterine benign lesion other than leiomyoma		39			9	
Other (including normal)		22			2	

BOT: borderline tumor; HGSC: high-grade serous carcinoma; LGSC: low-grade serous carcinoma.

**Table 3 cancers-14-00987-t003:** Sensitivity, specificity, and area under the receiver operating characteristic curve of the convolutional neural network.

Sequence	Interpreter	Sensitivity	95% CI	Specificity	95% CI	Accuracy	95% CI	AUC	95% CI	*p*-Value for AUC (Versus CNN)
T2WI	CNN	0.77	0.68–0.84	0.85	0.76–0.91	0.81	0.72–0.87	0.83	0.74–0.91	-
	Reader 1	0.63	0.54–0.68	0.90	0.82–0.96	0.77	0.69–0.82	0.89	0.82–0.95	0.127
	Reader 2	0.85	0.77–0.91	0.87	0.79- 0.92	0.86	0.78–0.92	0.91	0.85–0.97	0.048 *
	Reader 3	0.75	0.66–0.82	0.85	0.76–0.91	0.80	0.71–0.86	0.88	0.81–0.94	0.305
DWI	CNN	0.85	0.77–0.91	0.85	0.77–0.90	0.85	0.77–0.91	0.88	0.81–0.95	-
	Reader 1	0.71	0.61–0.79	0.77	0.68–0.84	0.74	0.65–0.82	0.81	0.72–0.89	0.151
	Reader 2	0.65	0.55–0.73	0.71	0.62–0.79	0.68	0.58–0.76	0.74	0.64–0.84	0.004 *
	Reader 3	0.79	0.70–0.87	0.75	0.66–0.82	0.77	0.68–0.84	0.82	0.73–0.90	0.135
ADC map	CNN	0.85	0.76–0.92	0.77	0.69–0.83	0.81	0.72–0.87	0.89	0.83–0.96	-
	Reader 1	0.81	0.72–0.88	0.75	0.66–0.82	0.78	0.69–0.85	0.84	0.77–0.92	0.263
	Reader 2	0.92	0.83–0.97	0.36	0.29–0.41	0.63	0.55–0.68	0.79	0.70–0.88	0.023 *
	Reader 3	0.77	0.68–0.84	0.79	0.70–0.86	0.78	0.69–0.85	0.85	0.78–0.93	0.356
CE-T1WI	CNN	0.81	0.73–0.86	0.92	0.85–0.97	0.87	0.79–0.92	0.86	0.78–0.94	-
	Reader 1	0.73	0.65–0.78	0.92	0.85–0.97	0.83	0.75- 0.88	0.87	0.79–0.94	0.903
	Reader 2	0.75	0.66–0.82	0.83	0.74–0.89	0.79	0.70–0.86	0.85	0.77–0.92	0.730
	Reader 3	0.75	0.65–0.83	0.71	0.62–0.79	0.73	0.64–0.81	0.82	0.73–0.90	0.416

ADC: apparent diffusion coefficient; AUC: area under the receiver operating characteristic curve; CE-T1WI: contrast-enhanced fat-saturated T1-weighted imaging; CI: confidence interval; CNN: convolutional neural network; DWI: diffusion-weighted imaging; T2WI: T2-weighted imaging. * *p* < 0.05.

**Table 4 cancers-14-00987-t004:** Inter-observer agreement between the convolutional neural network and the radiologists.

Comparison	Interpreter	T2WI	DWI	ADC Map	CE-T1WI
CNN vs. radiologists	1	0.42	0.50	0.42	0.63
	2	0.50	0.46	0.17	0.55
	3	0.45	0.56	0.42	0.36
Between radiologists	1 vs. 2	0.58	0.60	0.45	0.63
	2 vs. 3	0.68	0.50	0.39	0.52
	1 vs. 3	0.77	0.58	0.84	0.64

ADC: apparent diffusion coefficient; CE-T1WI: contrast-enhanced fat-saturated T1-weighted imaging; CNN: convolutional neural network; DWI: diffusion-weighted imaging; T2WI: T2-weighted imaging.

## Data Availability

The data presented in this study are not publicly available due to the ethical considerations.

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
