# Peer review of "Diagnosing Ovarian Cancer on MRI: A Preliminary Study Comparing Deep Learning and Radiologist Assessments"

_cancers, 2022, doi:10.3390/cancers14040987_

Round 1
Reviewer 1 Report
General comments: This is a retrospective study designed with a deep learning model and compare deep learning with radiologist assessment for diagnosing ovarian carcinoma using MRI. The advantage are (1) try to resolve the problem of pre-operative evaluation of ovarian tumor (2) larger images to create the machine learning model (3) compared the senior radiologists for clinical validations. However, the critical methodology issue are (1) lack of information of stages and other ovarian cancer related characters, such as ascites presentation or carcinomatosis, etc. These characters would be make a critical judgement by a radiologist. (2) validation only by making a diagnosis using a single sequence and this is different from clinical practice. Thus these conditions make the bias between radiologist and machine learning model. In fact, the value of pre-operative image to identify ovarian tumor is early stage (stage I-II).
The specific comments:
(1) Abstract: the conclusion seems CNN provided a diagnostic performance "not inferior to" the radiologist is better than " eqivalent"
(2) Introduction: Magnetic resonance imaging (MRI) is a better radiologic method for differentiating between malignant and benign ovarian tumors. However, MRI may not be suitable for patients with massive ascites.
(3) Methodology:
a. lack of information of stages in patient characters
b. exclusion criteria: malignant ovarian epithelial tumors mixed with non-epithelial com-94 ponents. However, why not exclude benign tumor with teratoma (stromal tumor) and normal ovaries.
c. one slice indicating a solid component, a central slice for each ovarian
tumor containing no solid components, or an image showing a normal caudal ovary in the absence of an ovarian tumor, was selected to test is not equitable.
Results: Table 3. How is the sensitivity and specificity of combine whole sequencing?
Author Response
Reviewer 1
General comments: This is a retrospective study designed with a deep learning model and compare deep learning with radiologist assessment for diagnosing ovarian carcinoma using MRI. The advantage are (1) try to resolve the problem of pre-operative evaluation of ovarian tumor (2) larger images to create the machine learning model (3) compared the senior radiologists for clinical validations. However, the critical methodology issue are (1) lack of information of stages and other ovarian cancer related characters, such as ascites presentation or carcinomatosis, etc. These characters would be make a critical judgement by a radiologist. (2) validation only by making a diagnosis using a single sequence and this is different from clinical practice. Thus these conditions make the bias between radiologist and machine learning model. In fact, the value of pre-operative image to identify ovarian tumor is early stage (stage I-II).
The specific comments:
- Abstract: the conclusion seems CNN provided a diagnostic performance "not inferior to" the radiologist is better than " eqivalent"
Thank you, I have changed it to "not inferior" in Abstract, Simple Summary, and beginning of the Discussion.
Introduction: Magnetic resonance imaging (MRI) is a better radiologic method for differentiating between malignant and benign ovarian tumors. However, MRI may not be suitable for patients with massive ascites.
Thank you, we have made the changes as suggested.
(3) Methodology:
- lack of information of stages in patient characters
Tumor stage information has been added to the Table 2.
And we have added the following sentence in the Result,” Tumor stage was comprehensively determined based on the International Federation of Gynecology and Obstetrics (FIGO) 2014 classification using pretreatment MRI, CT, surgical, and pathological findings.” in Patients of the M&M.
“The high percentage of stageⅠ malignant tumors was due in part to the inclusion of borderline tumors.” in the Results.
- exclusion criteria: malignant ovarian epithelial tumors mixed with non-epithelial com-94 ponents. However, why not exclude benign tumor with teratoma (stromal tumor) and normal ovaries.
In order to explore the possibility of developing the study into a screening test for ovarian cancer in the future, we included mature teratoma, the most frequent ovarian tumor, uterine myoma, which are sometimes difficult to distinguish from ovarian tumors, and normal cases.
- one slice indicating a solid component, a central slice for each ovarian
tumor containing no solid components, or an image showing a normal caudal ovary in the absence of an ovarian tumor, was selected to test is not equitable.
Since the test was performed on a single image, the images that can be diagnosed with a single image were arbitrarily selected. Therefore, although there is a selection bias as described in the limitations, both CNN and radiologists evaluate images under the same conditions.
Results: Table 3. How is the sensitivity and specificity of combine whole sequencing?
Although we did not evaluate the combined whole sequence, we added a description of the percentage of cases with false negative and false positive results for all sequences in the Results, and added it’s interpretation to the Discussion.
We added following sentence in the Results, “There was one case each in which the CNN showed false negative and false positive in all sequences. Reader 1 had three cases with false negatives, while readers 2 and 3 had no cases with false negatives or false positives in all sequences.“
We added following sentence in the Discussion, “and the fact that there was only one false positive and negative in all sequences in this study is a result that can be expected to improve the diagnostic performance of combination imaging in the future.”
Thank you for your accurate and informative review.

Reviewer 2 Report
Although the coefficient of variation is within acceptable range, the variation in age must be considered. Although images 4 through 6 are of good quality, the interpretation of the margins and boundaries of the images can be controversial for the evaluator.
Despite the limitations, the findings are valid for the scientific community in the area.
Author Response
Reviewer 2
Although the coefficient of variation is within acceptable range, the variation in age must be considered. Although images 4 through 6 are of good quality, the interpretation of the margins and boundaries of the images can be controversial for the evaluator.
As you said, it would be difficult to assess the borders and margins with DWI and ADC maps alone, especially in Figure 6, "and as the radiologists, it was assumed to be the misdiagnosis due to the difficulty distinguishing the solid component of the ovarian tumor from the intestinal tract" was added to the description of Figure 6 in the Results.
Despite the limitations, the findings are valid for the scientific community in the area.
Thank you for your accurate and informative review.

Reviewer 3 Report
Summary
Saida et al. present a prospective study to compare the diagnostic accuracy of a convolutional neural network to distinguish malignant from benign ovarian lesions. The CNN diagnoses are compared to diagnoses from three radiologists and then to the pathologic diagnosis for confirmation of malignant or benign lesions. Importantly, the authors point out that this analysis uses full pelvic images rather than cropped images of the tumors, which could improve the applicability of this CNN for clinical diagnosis. Also, borderline tumors, which are difficult to distinguish from benign lesions, were included in the training and test sets. This is a well written preliminary study that suggests the potential of CNN to distinguish malignant and benign ovarian lesions.
Major points:
The Introduction could provide a better explanation of the criteria used in the O-RADS MRI scoring system, which has very good sensitivity and specificity for identifying risk of malignancy.
In the Discussion, the authors stress the importance of including borderline tumors in their training and test sets and I agree that this is an important aspect of the study. Yet, they do not report specifically on the diagnostic accuracy of the CNN in the subset of borderline tumors. This is a smaller set of cases, but the diagnostic accuracy for borderline tumors should still be reported in the Results and the implications should be included in the Discussion.
The authors need to clarify this sentence, line 261: “Figure 6 shows a case in which the CNN and the radiologists showed false negatives on the sequences other than DWI, which was considered to be a misdiagnosis due to misidentification for radiologist.” I don’t understand the second part of this sentence. All radiologists and the CNN diagnosed it as malignant with DWI. Please define what constitutes a misdiagnosis in this study in the Methods or Results.
There is also an interesting point made in the Discussion about the difference between this study and the clinical setting where the radiologist can examine all modalities to determine a diagnosis. In the clinical setting, radiologists likely make calls where they have to rely more heavily on one modality than the others to make a diagnosis. If all four modalities are included for each patient (this should be mentioned in Methods), then, the design of this study (diagnosis from individual modalities) may provide an opportunity to establish a rationale for how to combine imaging modalities for the best diagnostic accuracy. For instance, if a malignant lesion is detected in more than one modality, is it more likely to be a correct diagnosis of malignancy? How often is detection of a malignancy in only one modality a false positive (imaging artifact) versus a true positive (better resolution in one modality vs the others). How often do the CNN or the radiologists miss a malignant lesion in all four modalities or do they usually identify a confirmed malignant lesion in at least one modality? How does this compare to identifying a benign lesion?
Minor points:
Please mention if there is any loss of resolution with the conversion of the DICOM images to JPEG.
Line 354: “clinical information such as age and tumor makers;” should probably be “tumor markers”.
Author Response
Reviewer 3
Summary
Saida et al. present a prospective study to compare the diagnostic accuracy of a convolutional neural network to distinguish malignant from benign ovarian lesions. The CNN diagnoses are compared to diagnoses from three radiologists and then to the pathologic diagnosis for confirmation of malignant or benign lesions. Importantly, the authors point out that this analysis uses full pelvic images rather than cropped images of the tumors, which could improve the applicability of this CNN for clinical diagnosis. Also, borderline tumors, which are difficult to distinguish from benign lesions, were included in the training and test sets. This is a well written preliminary study that suggests the potential of CNN to distinguish malignant and benign ovarian lesions.
Major points:
The Introduction could provide a better explanation of the criteria used in the O-RADS MRI scoring system, which has very good sensitivity and specificity for identifying risk of malignancy.
An explanation of the criteria used in the O-RADS MRI scoring system has been added to the Introduction.
“O-RADS MRI scores range from 0 to 5, with 0 indicating an incomplete evaluation, 1 indicating a normal ovary, 2 indicating a pure cystic mass, pure fatty mass, or pure endometriotic cyst that is almost certainly benign, 3 indicating a low-risk: cystic tumor with no enhancing solid tissue, or a tumor containing solid tissue with a low-risk time-intensity curve on dynamic contrast study, 4 indicating intermediate-risk: a tumor containing solid tissue with an intermediate-risk time-intensity curve, and 5 indicating high-risk: a tumor containing solid tissue with a high-risk time-intensity curve.”
In the Discussion, the authors stress the importance of including borderline tumors in their training and test sets and I agree that this is an important aspect of the study. Yet, they do not report specifically on the diagnostic accuracy of the CNN in the subset of borderline tumors. This is a smaller set of cases, but the diagnostic accuracy for borderline tumors should still be reported in the Results and the implications should be included in the Discussion.
The number of cases of borderline tumors that were actually diagnosed as malignant in each sequence was added to the Results and its interpretation was included in the Discussion.
In the Results, the following sentence was added, “The CNN correctly diagnosed 28 of 37 carcinomas and 9 of 11 borderline tumors on T2WI, 31 of 37 carcinomas and 10 of 11 borderline tumors on DWI and ADC map, and 31 of 37 carcinomas and 8 of 11 borderline tumors on CE-T1WI as malignant tumors.”
In the Conclusion, the following sentence was added, “In addition, there was no inferiority in diagnostic performance even for borderline malignant tumors, which are generally difficult to diagnose as benign or malignant.”
The authors need to clarify this sentence, line 261: “Figure 6 shows a case in which the CNN and the radiologists showed false negatives on the sequences other than DWI, which was considered to be a misdiagnosis due to misidentification for radiologist.” I don’t understand the second part of this sentence. All radiologists and the CNN diagnosed it as malignant with DWI. Please define what constitutes a misdiagnosis in this study in the Methods or Results.
We rewrote to “and as for the radiologists, it was assumed that the misdiagnosis was due to the difficulty in distinguishing the solid component of the ovarian tumor from the intestinal tract.“
There is also an interesting point made in the Discussion about the difference between this study and the clinical setting where the radiologist can examine all modalities to determine a diagnosis. In the clinical setting, radiologists likely make calls where they have to rely more heavily on one modality than the others to make a diagnosis. If all four modalities are included for each patient (this should be mentioned in Methods), then, the design of this study (diagnosis from individual modalities) may provide an opportunity to establish a rationale for how to combine imaging modalities for the best diagnostic accuracy. For instance, if a malignant lesion is detected in more than one modality, is it more likely to be a correct diagnosis of malignancy? How often is detection of a malignancy in only one modality a false positive (imaging artifact) versus a true positive (better resolution in one modality vs the others). How often do the CNN or the radiologists miss a malignant lesion in all four modalities or do they usually identify a confirmed malignant lesion in at least one modality? How does this compare to identifying a benign lesion?
When the respective diagnoses in each sequence of the same case were verified, there was one case each in which the CNN showed false negative and false positive in all sequences. Reader 1 had three cases with false negatives in all sequences, while readers 2 and 3 had no cases with false negatives or false positives in all sequences.
We added the following sentence in the Results, “There was one case each in which the CNN showed false negative and false positive in all sequences. Reader 1 had three cases with false negatives, while readers 2 and 3 had no cases with false negatives or false positives in all sequences.“
We added the following sentence in the Discussion, “and the fact that there was only one false positive and negative in all sequences in this study is a result that can be expected to improve the diagnostic performance of combination imaging in the future.”
Minor points:
Please mention if there is any loss of resolution with the conversion of the DICOM images to JPEG.
The JPEG format uses a lossless compression scheme.
We added the following sentence in the Dataset of the M&M, “which employs a lossless compression method.”
Line 354: “clinical information such as age and tumor makers;” should probably be “tumor markers”.
Thank you, we have changed.
Thank you again for your accurate and informative review.

Reviewer 4 Report
First of all, congratulations to the authors for their well-written and nicely designed work. Several comments:
- Ultrasound is more frequently used to diagnose ovarian lesions. Did the authors consider to perform a study using ultrasonography instead of MRI? If not, it would be worthwhile to do so.
- It is unclear why the authors decided to include uterine leiomyomas and benign uterine lesions in their analyses, as their objective was to make a diagnosis on ovarian lesions using deep-learning rather than differentiating uterine lesions. If their objective was to accurately differentiate other pelvic lesions from ovarian tumors, why did the authors then decide to exclude malignant lesions in the pelvis? Please explain this in more detail.
- For this study only single sequences were assessed, whereas for clinical purposes radiologists usually base their judgment on multiple sequences. Do the authors expect that CNN would still provide a diagnostic performance equivalent to that of the radiologists if multiple sequences were used by the radiologists, which is basically more representative of reality? Please explain.
- Line 161: 'the patients' (were blinded to the pathological and clinical findings) should be substituted by 'the radiologists' (....).
- Table 2: the abbreviation 'BOT' for borderline ovarian tumor is far more common than the abbreviation 'BLT', as used by the authors. Please modify.
Author Response
Reviewer 4
First of all, congratulations to the authors for their well-written and nicely designed work. Several comments:
Thank you so much.
- Ultrasound is more frequently used to diagnose ovarian lesions. Did the authors consider to perform a study using ultrasonography instead of MRI? If not, it would be worthwhile to do so.
- In our country, radiologists do not perform transvaginal ultrasound directly, so a study using ultrasound was not conceived.
- It is unclear why the authors decided to include uterine leiomyomas and benign uterine lesions in their analyses, as their objective was to make a diagnosis on ovarian lesions using deep-learning rather than differentiating uterine lesions. If their objective was to accurately differentiate other pelvic lesions from ovarian tumors, why did the authors then decide to exclude malignant lesions in the pelvis? Please explain this in more detail.
- In order to explore the possibility of developing the study into a screening test for ovarian cancer in the future, we included mature teratoma, the most frequent ovarian tumor, uterine myoma, which are sometimes difficult to distinguish from ovarian tumors, and normal cases. In this study, we excluded pelvic malignancies other than ovarian cancer because CNN cannot be used for screening if it can detect ovarian cancer but cannot detect other pelvic malignancies such as colon cancer and uterine cancer.
- For this study only single sequences were assessed, whereas for clinical purposes radiologists usually base their judgment on multiple sequences. Do the authors expect that CNN would still provide a diagnostic performance equivalent to that of the radiologists if multiple sequences were used by the radiologists, which is basically more representative of reality? Please explain.
We believe that if more images are available, better diagnostic performance can be. obtained with combined images. Furthermore, if we can optimize the weighting of each sequence, which radiologists do naturally, we can expect further improvement in diagnostic performance.
We added the following sentence in the Discussion, “and the fact that there was only one false positive and negative in all sequences in this study is a result that can be expected to improve the diagnostic performance of combination imaging in the future.”
- Line 161: 'the patients' (were blinded to the pathological and clinical findings) should be substituted by 'the radiologists' (....).
Thank you, we have changed.
- Table 2: the abbreviation 'BOT' for borderline ovarian tumor is far more common than the abbreviation 'BLT', as used by the authors. Please modify.
Thank you, we have changed.
Thank you again for your accurate and informative review.

Round 2
Reviewer 1 Report
All the questions are well response.
Reviewer 3 Report
The revised manuscript is sufficiently improved to warrant publication.